# Potential New Therapeutic Approaches Based on *Punica granatum* Fruits Compared to Synthetic Anthelmintics for the Sustainable Control of Gastrointestinal Nematodes in Sheep

**DOI:** 10.3390/ani12202883

**Published:** 2022-10-21

**Authors:** Fabio Castagna, Roberto Bava, Vincenzo Musolino, Cristian Piras, Antonio Cardamone, Cristina Carresi, Carmine Lupia, Antonio Bosco, Laura Rinaldi, Giuseppe Cringoli, Ernesto Palma, Vincenzo Musella, Domenico Britti

**Affiliations:** 1Department of Health Sciences, University of Catanzaro Magna Græcia, 88100 Catanzaro, Italy; 2Interdepartmental Center Veterinary Service for Human and Animal Health, CISVetSUA, University of Catanzaro Magna Græcia, 88100 Catanzaro, Italy; 3Pharmaceutical Biology Laboratory, Department of Health Sciences, Institute of Research for Food Safety & Health (IRC-FISH), University of Catanzaro Magna Græcia, 88100 Catanzaro, Italy; 4Department of Health Sciences, Institute of Research for Food Safety and Health IRC-FSH, University Magna Graecia of Catanzaro, 88100 Catanzaro, Italy; 5National Ethnobotanical Conservatory, Castelluccio Superiore, 85040 Potenza, Italy; 6Mediterranean Ethnobotanical Conservatory, Sersale (CZ), 88054 Catanzaro, Italy; 7Department of Veterinary Medicine and Animal Production, University of Naples Federico II, CREMOPAR, Regione Campania, 80100 Naples, Italy; 8Nutramed S.c.a.r.l., Complesso Ninì Barbieri, Roccelletta di Borgia, 88021 Catanzaro, Italy

**Keywords:** *Punica granatum*, Albendazole, Ivermectin, gastrointestinal nematodes, anthelmintic efficacy, sheep, animal welfare and health, green veterinary pharmacology

## Abstract

**Simple Summary:**

The spread of helminths resistant to the most common classes of anthelmintic drugs in sheep and the presence of drug residues in the environment has prompted research to find sustainable alternative solutions for gastrointestinal nematodes control. This study reports the results of in vivo studies on the efficacy of an aqueous *Punica granatum*-based macerate, used in southern Italy for the control of gastrointestinal nematodes in sheep. The anthelmintic efficacy was evaluated in parallel in sheep infected with gastrointestinal nematodes, using, as a positive control, the treatment with Ivermectin and Albendazole. The results of the study show a good anthelmintic efficacy, suggesting the use of this ethnoveterinary macerate as an alternative and sustainable therapeutical remedy for the helminthiasis control in the sheep.

**Abstract:**

Parasites, in particular, gastrointestinal nematodes (GINs) represent one of the main burdens affecting small ruminant farming and pose a serious threat to their health, welfare, productivity, and reproduction. The correct management of animals and the correct use of anthelmintic drugs are the pillars of the GIN control programs for small ruminants. However, globally due to the indiscriminate use of synthetic anthelmintics, there is a significant increase in anthelmintic resistance phenomena to one or more classes of drugs. Even if such a problem never represented a serious threat in southern Italy because of the favourable environmental conditions and because of the good farm management, the phenomenon is actually showing a steep increasing trend and requires alternative treatment measures and constant monitoring. The use of phytotherapies is considered a valuable alternative approach for GIN control in small ruminants and could help with reducing the amount of synthetic drugs used and the forthcoming anthelmintic resistance. From this perspective, the Calabria territory offers a wide number of plants with anthelmintic efficacy that could be helpful for this purpose. The aim of this study was to evaluate the anthelmintic efficacy of aqueous pomegranate (*Punica granatum* L.) macerate compared to the treatment with Ivermectin and Albendazole in sheep naturally infected with GINs. The pomegranate macerate derives from the ethnoveterinary knowledge of the Calabria region, Southern Italy. The anthelmintic efficacy was evaluated according to the faecal egg count reduction test (FECRt) using the FLOTAC techniques in two sheep farms in Southern Italy. The FECR was calculated from individual samples using the formula FECR = 100 × (1 − [T2/C2]). The treatment with Albendazole in the first farm showed an efficacy of 99.8% after 14 days and 94.8% after 21 days, while the treatment with Ivermectin in the second farm showed an efficacy of 99.9% after 14 days and 96.5% after 21 days of treatment. The pomegranate macerate, in both farms, showed a value of efficacy of around 50% from day 7 to day 21 after the treatment. Previous studies highlighted the presence of gallic acid as the main component in the pomegranate macerate, and its efficacy in nematode control has been as well previously demonstrated in other plant extracts. This in vivo study demonstrated the unequivocal efficacy of plant macerate in easily reducing 50% of the number of GIN eggs in sheep faeces. These results, obtained without the use of synthetic anthelmintics, indicate the use of green veterinary pharmacology as a sustainable alternative to the use of synthetic drugs to reduce the increase in drug resistance phenomena and the environmental impact.

## 1. Introduction

Parasites and parasitic diseases represent among the main challenges in health management in extensive sheep farming and are the cause of significant economic losses worldwide [1]. The costs are directly linked to the reduction in well-being and productivity, increased mortality, the widespread use of anthelmintics and higher managerial costs [2]. Traditionally, the therapeutic approach consists of the prescription and administration of chemotherapy, only rarely preceded by a visit to the animals and parasitological investigations [3,4]. This procedure, not in accordance with good clinical and preventive practice, is very problematic. The human intervention aims at a drastic reduction of the parasitic load, and for this purpose, it uses a high quantity of chemicals, which have serious environmental and public health repercussions. On the other hand, most of the products on the market for the treatment of gastrointestinal nematode (GIN) infection in farms are based on synthetic drugs [5,6]; they are generally broad-spectrum molecules that are administered to animals two to four times a year. The negative side effects of these drugs are known and studied. Synthetic anthelmintics have several other drawbacks, such as the risk of contamination by residues for food products, the negative effect of preventive treatments on the development of natural immunity against parasites, but the most important is the onset of parasite resistance phenomena towards the molecules used [7,8,9,10,11,12,13].

The most frequently reported resistance phenomena to refer to Benzimidazoles (BZ), such as Albendazole (ALB) and macrocyclic lactones (ML), in particular Ivermectin (IVM) [4]. Farm-level anthelmintic resistance (AR) in sheep and goats in Europe has been documented to reach 86% in BZ and 52% in ML. Some AR reports have also been documented on doramectin, moxidectin and monepantel [14,15,16]. Multi-resistance is also a documented growing phenomenon in France, Italy, and Greece [10] and is rapidly emerging at worrying levels in Scotland and France [11,12,13].

The phenomenon must raise the alarm for the small ruminant industry, especially in tropical and subtropical areas. In developing countries, natural nutritional resources are inadequate, and natural immunity is consequently compromised [17]. The wormers exert their action in different ways. However, within the same class, all products share the same mode of action. Therefore, when resistance arises to one product within a class, other products belonging to the same class are negatively affected. Synthetic pharmaceutical treatment must therefore be limited and, when carried out, must be administered in a targeted manner. The appropriateness of the antiparasitic treatment must be evaluated by the veterinarian in relation to the state of health of the animals, the quality, and the parasitic load present (quantitative parasitological analyses). In any case, the increasingly reported AR forces us to look for alternative substances to chemical treatments [6,18]. Furthermore, this crisis situation must push many farmers and veterinarians to deal with preventive medicine, such as integrated management against parasites [19]. A combination of anthelmintic administration and alternative methods of parasite control must be taken into account. From this point of view, medicinal plants have been used for pharmaceutical purposes since ancient times. However, their uses always remained in the customs of confined populations and the potential of such remedies was never fully explored, nor were non-compounds/mixtures opened. For example, chenopodium oil which derives from *Chenopodium ambrosioides,* has been used for many years in the UK for the treatment of parasitic nematode infections (*Strongylus, Parascaris* and *Ascaris* spp.) in monogastric animals, including humans [20]. Since the beginning of the last century, some parts of plants, such as dried leaves and flowers, have also been used as anthelmintics. In particular, Chenopodium is used for this purpose in Latin America [21]. Other species, including the male fern *Dryopteris filix-mas* and *Artemisia* spp., have shown efficacy against some cestodes of the genus *Moniezia* of ruminants and some nematodes of the genus *Ascaridia* spp. in poultry [22,23]. Recent investigations have identified many plants that could be destined or have the potential to be used as anthelmintics. The current shortage of chemotherapeutic agents, which can also be linked to resistance mechanisms, is pushing the scientific community to re-explore ethnopharmacological traditions to discover alternative drugs and remedies. Liu et al. [24] reported, in a review article, a summary of compounds of botanical origin published since 2002. The review of a good number of compounds that specify the plant of origin, efficacy against the parasitic model, the class of the molecule and the possible mechanism of action. However, only a few of those listed have been examined in vivo. In 2009, Garcia-Bustos et al. [25] published an extensive review that many natural products act against parasitic elements of humans and other animals. This review provides a good classification and description of the most important compounds in relation to the chemical structure.

Considering the botanical biodiversity, the Calabria region (southern Italy) offers a large panel of plants together with an extremely rich ethnobotanical and ethnopharmacological history. Botanical biodiversity is linked to the high geographical heterogeneity of the territory. The great difference in height between Southern Italy and the centre of the Mediterranean has created the ideal environment for the creation of a huge number of ecological niches capable of hosting such biodiversity. This heterogeneity has made possible the colonization of the territory by non-native plants, which, over the generations, have been incorporated into the ethnopharmacological knowledge of the territory [26]. Among these, *Punica granatum* is known to have been domesticated as early as the fifth millennium BC in the Mediterranean area. The pomegranate is known to have several constituents such as gallic acid, ellagic acid, phenolic punicalagins; and other fatty acids; catechin, rutin, quercetin and other flavonols; flavonones, flavones; tannins; anthocyanidins; and flavone glycosides. Among these, the constituents of the pomegranate that most have anthelmintic effects are alkaloids and tannins, although the synergistic action of the components of the phytocomplex must always be considered [27]. In other studies, parasiticidal activity has also been highlighted in vitro [28,29], but there are few in vivo studies that attest to its efficacy. This study describes the application of *P. granatum* as an ethnoveterinary remedy for the control of GINs in small ruminants; its anthelmintic activity is compared with the efficacy of the two most widely used IVM and ALB drugs.

## 2. Materials and Methods

### 2.1. Study Area and Animals

The ethics committee of the University of Catanzaro “Magna Græcia” has expressed a favourable opinion of this experimentation and all the experimental procedures performed on animals, with approval No. 97 of 09/10/2015.

This research was conducted in southern Italy (Calabria region), in an area with a typical Mediterranean climate. In this region, extensive farming is still very widespread. With 7.018 sheep farms and 219.368 sheep, on a national scale, the region occupies the fourth position for the number of farms and the fifth position for the number of sheep. These data, updated as of 31 December 2020, were provided by the National Database (BDN) of the Zootechnical Registry-CSN of the “G. Caporale” Institute of Teramo (Italy). Small ruminant farms represent a significant economic resource for the Calabrian agro-food industry, in particular for the dairy products sector [30].

Two semi-extensive sheep farms, with animals raised on hilly pastures (mean altitude 390 mt asl), were enrolled in this research, which was conducted between February and March. A farm screening was conducted 21 days prior to the trials (D-21). During this time, the farming system and the reared sheep were analyzed. On the first farm, dairy sheep were raised; on the second sheep for meat. In both farms, a breeding system with grazing animals was practised. The size of the flocks was comparable: 140 sheep, respectively, of the Comisana breed in the first herd (SF1) and of local mestizos in the second herd (SF2). The animals enrolled were 2 years of age (± 0.5) and had a live weight of 42 kg (± 1.8). All the animals had not received anthelmintic treatments in the past six months.

### 2.2. Ethno-Veterinary Remedy and Anthelmintic Drugs

In the two farms, the anthelmintic in vivo efficacy of an aqueous vegetable macerate, employed in local customs for the management of GINs in sheep was assessed. Its effectiveness was compared with two commonly administered anthelmintics drugs. The traditional mixture used in this study was a phyto-complex obtained from maceration in the water of ripe fruits (fruits and fruit rind) of pomegranates *(P. granatum).*

Drs. V. Musolino and C. Lupia, respectively, from the Department of Health Sciences at the University “Magna Graecia” of Catanzaro and Mediterranean Ethnobotanical Conservatory of Sersale (CZ), verified the taxonomic identification. With the access number *P. granatum*: 114, it is possible to consult the voucher specimen deposited at the Mediterranean Ethnobotanical Conservatory of Sersale (CZ), Italy. An elderly Calabrian breeder made the preparation used in the study by macerating local ripe pomegranates according to centuries-old traditions that are handed down from generation to generation. *P. granatum* fruits were harvested in October, at the peak of ripeness, in the province of Catanzaro, Calabria region of southern Italy, in an area at 800 m asl. Twenty kilograms of ripe pomegranate fruit, including the peel, were used to make the macerate. The ripe pomegranates were divided into four pieces and macerated in 60 L of previously boiled spring water. An old Calabrian farmer made this macerate by macerating the ripe pomegranates in spring water for at least 10 months, including the fruit and the peel [31]. The macerate was then filtered using a cotton filter after that. Approximately 70% of the whole beginning amount was the average yield. The dosages recommended by the breeder who made and employed the aqueous *P. granatum* macerate over the years were used.

The anthelmintics drugs used were ALB drug-based and IVM drug-based. The anthelmintics chosen for this study belong to the groups of Benzimidazoles and Macrocyclic lactones, the drugs currently most used in Europe for GIN control in small ruminants [2]. These drugs were given at the dosages recommended by the manufacturers for the stated purposes, with the utmost consideration for animal welfare. Aqueous pomegranate macerate was used in tests on groups of sheep (15 sheep per group), with a single oral dosage of 50 mL, comparing its effectiveness with ALB at 3,75 mg/kg/BW/orally and IVM at 200 μg/Kg B/W/subcutaneously, at a single dose, respectively in SF1 and SF2.

At day-7 in the two sheep farms (SF 1 and 2), a parasitological screening was done, taking faecal samples from 180 sheep (90 animals/sheep farm) for subsequent copromicroscopic examinations. At day-7 in the two farms, a parasitological screening was done, taking faecal samples from 180 sheep (90 animals/SF) for subsequent copromicroscopic examinations. Based on the results of this screening, sheep for the experimental groups (45 animals per farm), homogeneous in terms of parasite intensity, were selected. Each group consisted of 15 animals, and the different study groups were assigned the following acronyms: SF1: PG1 (*P. granatum* group 1), treated with aqueous *P. granatum* macerate (50 mL/sheep/orally/single administration); ALB, treated with ALB drug (3.75 mg/kg/BW/orally/single administration); CG1, untreated.

SF2: PG2 (*P. granatum* group 2), treated with aqueous *P. granatum* macerate (50 mL/sheep/orally/single administration); IVM, treated with IVM drug (200 μg/Kg B/W/subcutaneously/single administration); CG2, untreated.

Animals were grouped on day zero (D0), and faeces samples were taken to perform the faecal egg count (FEC). Following that, the anthelmintic treatments were administered to the PG1, PG2, ALB, and IVM groups. Faeces were tested for FEC on day 7 (D7). At days 14 (D14) and 21 (D21), faecal samples were collected, and the faecal egg count reduction (FECR) was calculated to assess the effectiveness of the anthelmintic.

### 2.3. Chemical Characterization

Aqueous *P. granatum* macerate used in this study is an aliquot that was previously used and characterized in our previous in vitro tests [31]. Using liquid chromatography-electrospray ionization mass spectrometry (LC/MS-ESI), this macerate was examined. Thermo Scientific’s (Rodano, MI, Italy) Dionex Ultimate 3000 RS was used for the chromatography. A Thermo Scientific Q-Exactive (Rodano, MI, Italy) mass spectrometer was used for High Resolution Mass Spectrometry (HRMS) [31].

### 2.4. Parasitological Analysis and Anthelmintic Efficacy

Individual FEC were calculated with the FLOTAC technique using a sodium chloride-based flotation solution with a specific gravity of 1.200 (FS2), the “gold standard” for GIN egg counts (detection limit = 2 eggs per gram (EPG) of faeces) [32].

Additionally, a pooled faecal culture was carried out for each group at D0 in accordance with the Ministry of Agriculture, Fisheries, and Food’s procedure (MAFF) [33]. Utilizing the morphological keys suggested by van Wyk and Mayhew, developed third-stage larvae (L3) were identified [34].

One hundred L3 were used for identification and percentages of each nematode genus; if less than 100 L3 were present, all larvae were recognized. It was, therefore, able to calculate the proportion of each species based on the total number of larvae detected.

Anthelmintic efficacy was evaluated using the FECR test, according to the World Association for the Advancement of Veterinary Parasitology (WAAVP) guidelines (Coles et al., 1992) [35]. In particular, to evaluate the anthelmintic efficacy, the arithmetic mean of the EPG for the faecal samples of the study groups (PG1, ALB, CG1 and PG2, IVM, CG2) was calculated and for each treatment group (PG1, ALB and PG2, IVM) the percentage efficacy (%) was calculated in terms of FECR in the different days (D7, D14, D21). Based on the arithmetic mean of the untreated and treated groups, the following equation was used to determine the anthelmintic efficacy: FECR = 100×(1-[T2/C2]).

The mean post-treatment FEC of the treated group is represented by T2 in the formula above, while the mean post-treatment FEC of the untreated control group is represented by C2 [36,37].

### 2.5. Statistical Analysis

Data were analyzed with GraphPad PRISM 9.1.2 (version 9.1.2). The results are expressed as mean ± S.D. Normality was tested using Shapiro–Wilk test. Data without normal distribution were analyzed using Kruskal–Wallis analysis of variance followed by Dunn’s tests. The Mann-Whitney test was used for the comparison of data derived from two specific groups.

A repeated measures ANOVA was performed to compare the treatments at the experimental times of observation. The sphericity assumption was checked by Mauchly’s test, then, a Greenhouse-Geisser correction was performed to adjust the violation of the sphericity assumption. Tukey’s post hoc test was performed for pairwise comparisons. The repeated measures ANOVA was performed with JASP (version 0.16.3).

Value with *p* < 0.05 were considered statistically significant.

## 3. Results

### 3.1. Parasitological Studies and Anthelmintic Efficacy

*P. granatum* macerate was tested in two different and independent trials (SF1 and SF2, respectively). Every trial included three different experimental groups: the negative control group (CG1 and CG2, control, animals treated with a placebo), the positive control group (ALB or IVM) and the group treated with *P. granatum* macerate (PG1 and PG2).

Although it is not scientifically relevant to make a comparison of the efficacy of *P. granatum* with ALB or IVM 7 days after the treatment, it is important to highlight that at this time, PG1 and PG2 groups showed an FECR of 55.7% (*p* < 0.01, Table 1, Figure 1) and 56.2% (*p* < 0.01, Table 2, Figure 2), while ALB and IVM groups showed an FECR of 83.5% (*p* < 0.001; Table 1 and Figure 1) and 73.6% (*p* < 0.05; Table 2 and Figure 2), respectively.

In SF1 trial, the ALB treatment reduced up to 99.8% the FECR after 14 days (*p* < 0.001) and this reduction persisted up to 94.8% after 21 days (*p* < 0.001). The parallel treatment with *P**. granatum* showed a reduction of 53.9% (*p* < 0.05) after 14 days, persisting up to 46.1% at day 21 (*p* < 0.001; Table 1 and Figure 1).

Likewise, in SF2 trial, the IVM treatment reduced up to 99.9% (*p* < 0.001) the FECR after 14 days and the reduction persisted up to 96.5% (*p* < 0.001) after 21 days. At the same time, the treatment with *P. granatum* showed a reduction of 54.2% (*p* < 0.01) after 14 days from the treatment persisting up to 45.7% at day 21 (*p* < 0.01; Table 2 and Figure 2).

All the collected samples were qualitatively analyzed for the nematode’s composition. The genera of nematodes detected in both farms, in all groups and on all days of sampling were *Haemonchus, Trichostrongylus, Teladorsagia* and *Chabertia,* as shown in Table 3 and Table 4.

The results of the repeated measures ANOVA showed that there was a statistically significant main effect on the sheep third-stage larvae nematode percentages, both for SF1 (F(2.650,11.925) = 4.557, *p* < 0.05) and SF2 (F(2.704,12.170) = 5.344, *p* < 0.05).

However, *P. granatum* macerate treatment didn’t significantly reduce the sheep nematode third-stage larvae percentages in both trials.

### 3.2. Chemical Characterization

The macerate was fractionated according to the solubility in methanol. The two obtained fractions were analyzed by LC-HRMS. The analysis returned several peaks of phenolic acids and ellagitannins. To fully identify them, an MS/MS ESI study was carried out. Table 5 displays the characterisation, including the m/z values (obtained for LC/HRMS, ESI(-)) of each component and their structural identification [31].

## 4. Discussion

Ethnoveterinary remedies are poorly documented everywhere in the globe, and, most of the knowledge is threatened by the absence of proper track records necessary to guarantee survival during the time of the high number of known remedies. In recent years, also due to the widespread phenomena of drug resistance reported, natural remedies are increasingly of interest to the scientific community. There are several phytotherapics tested in laboratory and field tests in different animal species [38,39,40]. The results have often been encouraging, so it is necessary to continue the path of enhancing the knowledge linked to the science of ethnoveterinary medicine. A decrease in the use of the synthetic drug must necessarily pass from its association with natural remedies, which are useful for keeping the parasitic charge below the damage threshold. Only with an integrated approach of this type can the increasingly widespread trend towards the development of drug resistance be reduced [41]. The reduction of the synthetic drug is also important in light of another consideration: “medicinal products have a particular environmental impact”. As an example, we can talk about avermectins which are the most used category [42]. These molecules are excreted, to a certain extent, with the faeces of treated animals and have a long persistence in the environment [43,44,45]. Once dispersed, they are toxic to many species of invertebrates, which play a fundamental role in maintaining a balance of both the aquatic and terrestrial ecosystems. By way of example, it can be said that they also belong to the orders of *Anoplura, Dictyoptera, Homoptera, Thysanoptera, Colaptera, Siphonaptera, Diptera, Lepidoptera and Hymenoptera.* The species in these orders are essential for maintaining pasture hygiene, nutrient cycle, soil aeration, humus content, water percolation and pasture productivity. Furthermore, they also ensure that the grazing area of livestock is not drastically reduced by an accumulation of dung. In the cow dung community, dung-eating flies, coprophagous coleopters and annelid worms are the most important organisms [43,46]; they are also a source of nutritional elements for vertebrate animals, such as birds, amphibians, and mammals. A wide use of avermectins tends to decrease biodiversity with a strong environmental impact; moreover, in light of the above, the use of the synthetic drug must be reduced in the farm. This reduction can only help to trigger a virtuous circle for a new conception of animal husbandry.

In this described work, the anthelmintic efficacy of *P. granatum* macerate was evaluated in two different trials in comparison with the two most used anthelmintic drugs IVM and ALB. As in Figure 1a, both trials started with the creation of three homogenous groups according to the timepoint zero (D0) FEC. FEC was then measured for all three groups on day 14 and day 21 after the scheduled treatments. It needs to be mentioned that *P. granatum* macerate efficacy was consistent even at 7 days after the treatment. In our studio, the effects of the treatment revealed that the aqueous pomegranate extract reduces the EPG of gastrointestinal nematodes in treated subjects compared to the control group. In particular, we recorded an FECR of 46.1% in the group compared with Albendazole and 45.7 in the group compared with Ivermectin. IVM and ALB drugs clearly showed higher efficacy, not demonstrating AR phenomena.

According to the coprocultre data, four GIN genera—*Haemonchus, Trichostrongylus, Teladorsagia,* and *Chabertia*—were present on the farms that were under investigation. The results from the cultures of all groups did not reveal any appreciable variation in the ratio between the percentage of genera detected previously and after treatment, despite minor alterations in their percentages being seen when comparing various treatment groups. This finding shows that none of the applied therapies are genus-specific. The *P. granatum* macerate was previously analyzed and showed the presence of compounds with high anthelmintic efficacy (single dose administration of 50mL). The anthelmintic efficacy is due to the composition mainly made of alkaloids, tannins, flavonoids, glycosides, and phenols. Among the last ones, the major part is composed of gallic and ellagic acid, as documented in previous studies concerning pomegranate parts [47]. This plant extracts have a demonstrated antiprotozoal [48,49,50], anticestodal [51], antinematodal [52] and antitrematodal [53] activity. These compounds and their efficacy are, as demonstrated, effective without the application of complex extraction techniques; it is indeed sufficient to collect the fruits and prepare a macerate at room temperature.

The results of the study showed a good anthelmintic efficacy of the pomegranate-based macerate, suggesting the use of this Calabrian ethnoveterinary preparation as an alternative and sustainable therapeutic remedy for the control of helminthiasis in sheep. The use of this preparation could thus gradually reduce the phenomena of resistance to anthelmintics and, at the same time, improve the welfare and health of the animals. In addition, considering that aqueous *P. granatum* macerate is completely natural, that the doses required are very low and that the plant easily grows in the Mediterranean area, this is certainly an easy-to-use and green method for GIN control in small ruminants. However, its effectiveness was lower than the Ivermectin and Albendazole compounds. Therefore, the introduction of the *P. granatum* mixture in integrated parasite management programs could be recommended. In this way, together with the rotation of drugs, there would be a substantial decrease in the use of the synthetic drug and the consequent possibility of developing drug resistance phenomena.

## 5. Conclusions

In this in vivo study, an aliquot of pomegranate macerate was used already tested in vitro test in 2020 by Castagna et al. [31]. There are many in vitro tests aimed at certifying the control efficacy of numerous plant species and their mixtures, but the field tests are much inferior. This discrepancy is not insignificant-the in vitro tests are performed in environments with controlled conditions where interference is minimized or mostly eliminated, and the results obtained are often not confirmed in the vivo tests. Therefore, this study is doubly valuable since, in addition to putting laboratory experience into practice, it compares in vivo the antiparasitic efficacy of an ethnoveterinary naturally remedy based on *P. granatum* with the two synthetic drugs most used in sheep farms.

The macerate of *P. granatum* returned particularly convincing efficacy results for a natural blend. The parasite control provided may allow consideration of its introduction into therapeutic control programs. In fact, together with good farm management, the rotation of pastures and the active ingredients used in sheep farms, it can help not only to improve animal welfare and health but would also help slow down the phenomena of anthelmintic resistance and the environmental impact of farms.

In conclusion, this study further develops the important branch of Green Veterinary Pharmacology, an area of veterinary medicine that should necessarily be implemented to make animal husbandry sustainable in a world continuously subjected to the action of environmental pollution, also deriving from the thoughtless use of drugs.

## Figures and Tables

**Figure 1 animals-12-02883-f001:**
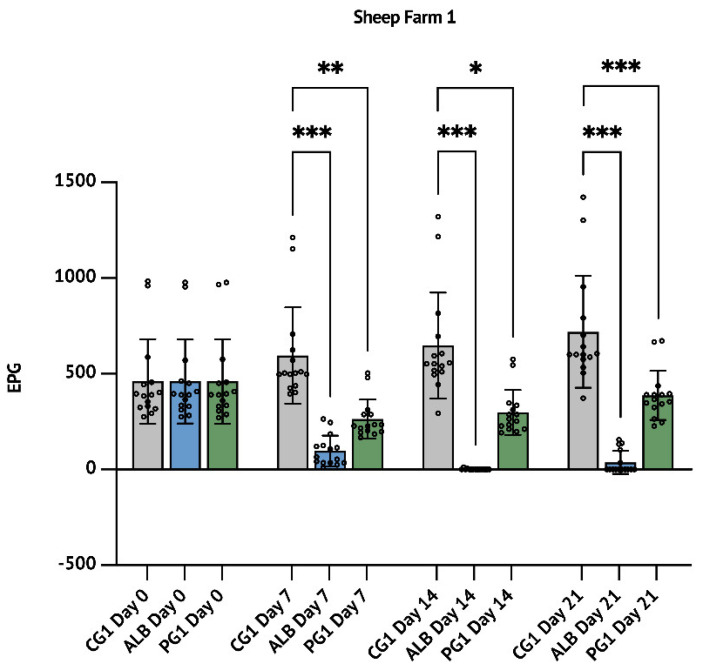
Sheep farm 1 (SF1): results concerning the comparison between the anthelmintic efficacy of the aqueous *P. granatum* macerate with Albendazole. The results are expressed as mean ± S.D. * *p* < 0.05, ** *p* < 0.01, *** *p* < 0.001.

**Figure 2 animals-12-02883-f002:**
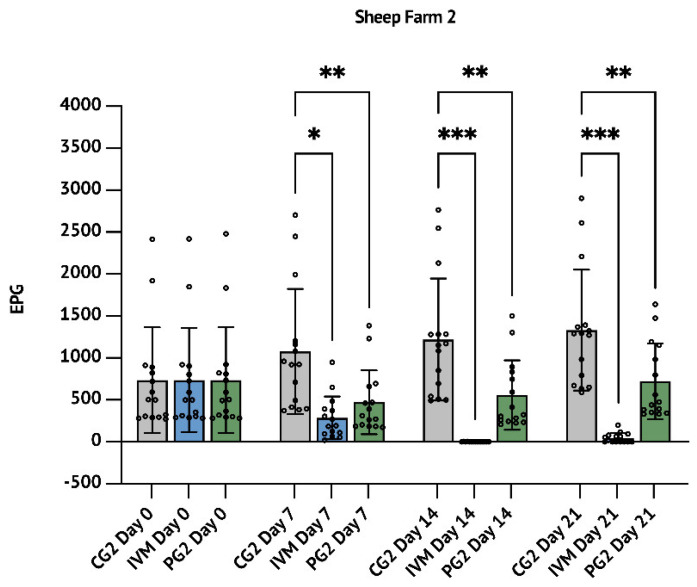
Sheep farm 2 (SF2): results concerning the comparison between the anthelmintic efficacy of aqueous *P. granatum* macerate with Ivermectin. The results are expressed as mean  ± S.D. * *p* < 0.05, ** *p* < 0.01, *** *p* < 0.001.

**Table 1 animals-12-02883-t001:** Sheep farm 1 (SF1): results concerning the comparison between the anthelmintic efficacy of the aqueous *P. granatum* macerate with Albendazole. PG1: treated aqueous *P. granatum* macerate group 1 (50/mL/OS as a single dose); ALB: treated Albendazole group (3.75 mg/Kg/BW/OS as a single dose); CG1: untreated control group 1; gastrointestinal nematodes (GINs) eggs per gram (EPG) of faeces (mean) of the different groups and the faecal egg count reduction (FECR) (%) at different days (D) after treatment; SD (standard deviation).

SF1 Groups	D_0_	D_7_		D_14_		D_21_	
	EPG Mean (SD)	EPG Mean (SD)	FECR %	EPG Mean (SD)	FECR %	EPG Mean (SD)	FECR %
PG1	460 (±221.4)	264 (±101)	55.7	298.8 (±117.1)	53.9	387.6 (±128.5)	46.1
ALB	460 (±218.7)	98 (±79.9)	83.5	1.2 (±3.4)	99.8	37.6 (±60.5)	94.8
CG1	460 (±221.6)	596 (±251.7)	-	647.6 (±277.2)	-	719.6 (±292.1)	-

**Table 2 animals-12-02883-t002:** Sheep farm 2 (SF2): results concerning the comparison between the anthelmintic efficacy of aqueous *P. granatum* macerate with Ivermectin. PG2: treated aqueous *P. granatum* macerate group 2 (50/mL/OS as a single dose); IVM: treated Ivermectin group (200 μg/Kg B/W/SC as a single dose); CG2: untreated control group 2; gastrointestinal nematodes (GINs) eggs per gram (EPG) of faeces (mean) of the different groups and the faecal egg count reduction (FECR) (%) at different days (D) after treatment; SD (standard deviation).

SF2 Groups	D_0_	D_7_		D_14_		D_21_	
	EPG Mean (SD)	EPG Mean (SD)	FECR %	EPG Mean (SD)	FECR %	EPG Mean (SD)	FECR %
PG2	735.2 (±628.9)	471.6 (±379.7)	56.2	557.9 (±414.6)	54.2	723 (±452.5)	45.7
IVM	735.7 (±620.6)	284.4 (±255.6)	73.6	0.8 (±2.1)	99.9	46 (±58.5)	96.5
CG2	735.7 (±631.5)	1077 (±747.4)	-	1219 (±728.7)	-	1332 (±719)	-

**Table 3 animals-12-02883-t003:** Percentage of sheep nematode third-stage larvae (L3) for each group at D0, D7, D14 and D21 in sheep farm 1 (SF1); PG1 treated aqueous *P. granatum* macerate group 1 (50/mL/OS as a single dose); ALB treated Albendazole group (3.75 mg/Kg/BW/OS as a single dose); CG1 untreated control group 1.

SF1 Groups	Day D	*Haemonchus* (%)	*Trichostrongylus* (%)	*Teladorsagia* (%)	*Chabertia* (%)
PG1	D0	13	45	36	6
	D7	18	38	32	12
	D14	11	32	43	14
	D21	9	35	48	8
ALB	D0	21	35	41	3
	D7	15	40	44	1
	D14	0	0	0	0
	D21	0	0	0	3
CG1	D0	18	32	42	8
	D7	11	30	45	14
	D14	14	28	47	11
	D21	8	35	44	13

**Table 4 animals-12-02883-t004:** Percentage of sheep nematode third-stage larvae (L3) for each group at D0, D7, D14 and D21 in sheep farm 2 (SF2); PG2 Table 2. (50/mL/OS as a single dose); IVM treated Ivermectin group (200 μg/Kg B/W/SC as a single dose); CG2 untreated control group 2.

SF2 groups	Day D	*Haemonchus* (%)	*Trichostrongylus* (%)	*Teladorsagia* (%)	*Chabertia* (%)
PG2	D0	29	32	38	1
	D7	25	27	44	4
	D14	27	21	47	5
	D21	32	24	44	0
IVM	D0	19	35	46	0
	D7	25	30	45	0
	D14	0	0	0	0
	D21	0	0	0	0
CG2	D0	31	25	43	1
	D7	34	20	46	0
	D14	29	26	45	0
	D21	39	19	41	1

**Table 5 animals-12-02883-t005:** Chemical characterization results (adapted from Castagna et al., 2020).

Peak LC-MS	*m*/*z* Theoretical	*m*/*z* Measured	Analyte and Molecular Formula
(1)	149.0092 181.0718 193.0354 481.0697	149.0081 181.0711 193.0347 481.0626	Tartaric acid (C_4_H_5_O_6_) Mannitol (C_6_H_13_0_6_) Glucuronic acid (C_9_ H_9_ O_7_) 2,3-(S)-hexahydroxyphenyl-D-glucose (C_20_H_17_O_14_)
(2)	169.0142	169.0134	Gallic acid (C_7_H_5_O_5_)
(3)	288.9990 469.0049	288.9992 469.0050	Phelligridin J (C_13_H_5_O_8_) Valoneic acid dilattone (C_21_H_9_O_13_)
(4)	197.0455	197.0449	Syringic acid (C_9_H_9_O_5_)
(5)	-	186.1129	unknown (C_13_H_14_O)
(6)	300.9990 447.0642	300.9991 447.0573	Ellagic acid (C_14_H_5_O_8_) Ducheside A (C_20_H_15_O_12_)

## Data Availability

The data are kept at the University of Magna Græcia of Catanzaro and are available upon request.

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
