# Peer review of "Potential New Therapeutic Approaches Based on Punica granatum Fruits Compared to Synthetic Anthelmintics for the Sustainable Control of Gastrointestinal Nematodes in Sheep"

_animals, 2022, doi:10.3390/ani12202883_

Round 1

Reviewer 1 Report (Previous Reviewer 2)

the paper is good. But in line 173, change the word effecacy  to efficacy

Author Response

Response to Reviewer 1 comment

The paper is good. But in line 173, change the word effecacy to efficacy

  1. Thank you very much for taking the time to review the article. Many thanks for the appreciation and for the suggestion. We have corrected the errors.

Reviewer 2 Report (Previous Reviewer 1)

Please find the suggestions attached.

Author Response

Response to Reviewer 2 comment

Comments Article ANIMALS Revised version 2 pomegranate Based on this revised version, it is clear that the authors have taken into account most of the comments on the initial version and the version R1 I have still a few comments but mainly on the form of the manuscript

1 Concerning the data on the percentage of third stage larvae Figure 3 is totally redundant with the tables 3 and 4. In addition, the figure 3 is more confusing than the tables (use of different scale for Y axis according to the nematode genus). Therefore, I suggest to delete the figure 3. In addition, the comments from line 335 to line 338, seem irrelevant since the main effect is the nearly total
reduction in EPG provoking of course a reduction of larvae. On the other hand, the comparison of the effects of pomegranate vs control data on the percentage of larval genera is important to keep (line 339-340)

R. Thank you very much for the comment and for the suggestion. We agree with you. We deleted Figure 3, the comments from line 335 to line 338, and kept the comparison of pomegranate effects versus the control data on the percentage of larval genera (line 339-340).

2 References list

This section should be checked throughly to respect the journal recommendations to verify teh following
- Use or not of capital letters for each word in the title ? (see for example, comparison between ref 5 and 6)

- Need to user italics according to the taxonomic rules for the name of species (see for example, line 506 (ref 18 ) or line 529 ref 28)

- Check the quality of journal quotation in a few references (eg ref 23 name of the journal ? ref 27 Proceedings of proceedings ? Ref 33 I think this is the manual of MAAF published by the UK government Please change the whole reference; Ref 47 please complete and check the journal refrence

R. Thanks so much for the comment and for the suggestion. We have carefully checked the section and, as indicated, we have corrected the capital letters of each word in the title, we have corrected the genus and species in italics and finally we have indicated the name of the magazine at ref 23, corrected at ref 24 and 33 and checked and completed ref 47.

Conclusion
This article is providing interesting results on the use of pomegranate’s macerate and the possible
anthelmintic properties against gastro intestinal nematodes in small ruminants.This corresponds to the concept of a « green », more sustainable therapeutic ways in animal production. The results are worth to be published after editorial revisions of the current version

R. Thank you very much for taking the time to the paper review. Many thanks for the suggestions and comments that have helped greatly improve our article.

This manuscript is a resubmission of an earlier submission. The following is a list of the peer review reports and author responses from that submission.

Round 1

Reviewer 1 Report

SEE MY COMMENTS SENT SEPARATELY  IN THE FILE BELOW

Author Response

Comments on the article « Potential new therapeutic approaches based on Punica granatum compared to synthetic anthelmintics for the sustainable control of gastro intestinal nematodes in sheep » submitted for publication to ANIMALS by F. Castagna and colleagues.

This article aims at confirming the traditional use as anthelmintic of macerate of pomegranate against gastro intestinal nematodes (GINs) of sheep. The manuscript is based on 2 in vivo studies in 2 distinct farms in Italy, based on well-designed protocols with 15 sheep per group. The subject of the study is within the scope of Animals

The manuscript is acceptable for publication after some major revisions whose main points are indicated herein
• The main objective of the study is briefly but clearly defined at the end of the introduction (line 139-141). However, the mode of presentation of the results is disturbing because the authors seem more focusing on results obtained with the positive control (synthetic AH either albendazole or ivermectin) and on the other hand, providing limited and diluted information on the potential effect of pomegranate macerate, which is the real original aim of the study.

  • To improve this last point, it would be useful
    - To describe clearly the results of comparison date by date of the EPG values in pomegranate vs control group in both farms

RE: Thank you for your suggestion. We have added the statistical analysis perform in the materials and methods section, the statistical results in the results section and we have discussed the statistical

results obtained in the discussions section.
- To present also the results of D7 for the 2 farms and 3 groups in figure 1

RE: Thank you very much for your suggestion which helped to improve manuscript. The information has been added.

- To complete Table 3 by providing an indication of the variability (e.g. Standard deviation) and to indicate clearly the results of the statistical analyses at one glance

RE: Thank you very much for your suggestion which helped to improve manuscript.

The information has been added.
- To question the use of 2 tables (1 and 2) presenting the main general recovered on the different dates. However, there is no statistical analyses provided to interpret these data

RE: Thank you very much for your suggestion. We prefer to keep two tables as the results refer to two different farms. We have added added the statistical analysis.
• Section 2.5 : Statistical analyses (line 242-244)
This section should be completed (eg to compare the different proportion of species genera described in Table 1 and 2)

RE: Thank you very much for your suggestion which helped to improve manuscript.

The information has been added.
Also, to reinforce their comparison, the authors might benefit to apply a time repeated ANOVA analysis which has the advantage to include the evolution of data obtained on the 3 dates of observation (D7, D14 and D21) to conclude whether or not the sheep receiving the pomegranate macerate has some statistical effects when compared to the control group.

RE: Thank you for your suggestion. We performed this new statistical analysis and included it in the test.
• The authors should check thoroughly in the whole manuscript for editioral issues and quality of . See for example :
- Line 28 reported ? Line 211 Check the spelling of characterization ; line 303 enrolled
?. Table 5 analySE instead of analyTE

RE: Thank you for your suggestion. The manuscript has been checked and the typos have been corrected. In table 5, the correct term is analyte, as it indicates the component, the chemical species or the chemical constituent deriving from the analytical procedure.

  • The authors should also verify the whole list of references to respect the taxonomic codified rules in regard of the name of species (use of italics) For example, see ref 18 , 28, 35,36, 46, 49 etc
    RE: Done

Reviewer 2 Report

Author

the suggest are in the file

thanks

Author Response

Rewiew about alternative therapheutic to helminths

Introduction
It is better to write something about action mechanism of the Punica granatum in introduction

RE: Thanks for your advice which helps improve the manuscript. We have added several concepts according to it in the introduction.
Line 29 – promegranate – It is better write the scientific name

RE:. Now amended according to the suggestion.
Line 31 - GIN – It is necessary to write to significate of abbreviate

RE: Now amended according to the suggestion.

Method
Line 192 – single dose – one time? Several days?. It is better rewrite and the sentence more clear.

RE: Now it is better explained.

Line 200 – What means PG1? Is it Punica granatum? Then, it is better write in parenthesis
RE: Done.

Line 203 – PG2? Similar to PG1
RE: Done.

Line 211 - Chemical charachterizatin - the process should write step to step.
RE: The chemical characterization is described in detail in the in vitro study, cited, conducted by Castagna et al, 2020. Please see the following bibliographical reference: Castagna, F.; Britti, D.; Oliverio, M.; Bosco, A.; Bonacci, S.; Iriti, G.; Ragusa, M.; Musolino, V.; Rinaldi, L.; Palma, E. In Vitro Anthelminthic Efficacy of Aqueous Pomegranate (Punica granatum L.) Extracts against Gastrointestinal Nematodes of Sheep. Pathogens 2020, 9, 1063.

Line 212 Aqueous Punica granatum macerate in 10 months? It should write the time
RE: Thanks a lot for the tip. We have rephrased the sentence and added the bibliographic reference. this part is described in detail in the in vitro study, cited, conducted by Castagna et al, 2020. Please see the following bibliographical reference: Castagna, F.; Britti, D.; Oliverio, M.; Bosco, A.; Bonacci, S.; Iriti, G.; Ragusa, M.; Musolino, V.; Rinaldi, L.; Palma, E. In Vitro Anthelminthic Efficacy of Aqueous Pomegranate (Punica granatum L.) Extracts against Gastrointestinal Nematodes of Sheep. Pathogens 2020, 9, 1063.

Line – 219 - FLOTAC – It should write the significate
RE: FLOTAC is a multivalent technique for qualitative and quantitative copromicroscopic diagnosis of parasites in animals and humans. For more information on the technique, see the following bibliographical reference: Cringoli, G.; Rinaldi, L.; Maurelli, M.P.; Utzinger, J. FLOTAC: new multivalent techniques for qualitative and quantitative opromicroscopic diagnosis of parasites in animals and humans. Nat. Protoc. 2010, 5, 503.

Line 286 it is important to emphasize that at 7 days after treatment 285 the PG (1 and 2) groups showed an FECR of 56.2% and 55.7%, respectively in SF1 and 286 SF2. This result is not in the table 1 or 2. It should be written in table.
RE: Thank you very much for your suggestion which helped to improve manuscript.

The information has been added.

Line 288 – It shoud write the panels are in figure 01. The reader looking for panel in text and the reader become lost.
RE: Done.

Line 291 – what figure? It is not cited In text
RE: Modified.

Why the figure 1 did not have date about D7?
RE: Thank you very much for your suggestion which helped to improve manuscript.

Has now been inserted.

Discussion
Line 327 – Other consideration? The author should write the consideration
RE: Thanks for the comment. The text has been changed to make it better understandable.

Line 349 - P. granatum macerate efficacy was consistent even at 7 days after the treatment. It is not in table. In day 14 and 21, the % efficacy is more high than IVM and ALB. This question shoud be explained.
RE: As can be seen in both the table and the graph, Punica granatum returned less efficacy than the synthetic drugs at day 7, 14 and 21. The missing information, regarding day 7, has been added.

In discussion, it is better to write compounds of ivermectin and albendazole. Because this antiparasite are more effectiveness then Punica granatum. It should necessary the theme be more discussed.
RE: Thanks for your comment which helps improve the manuscript. A paragraph has been added in accordance with the suggestion.

Conclusion
Line 373 – the conclusion should be rewrite in papper. It didn not show to reader the conclusion the papper

RE: The conclusions have now been written in accordance with the suggestion.

Round 2

Reviewer 1 Report

Based on this revised version, it is clear that the authors have taken into account most of the comments on the initial version and tried to send a revised version rapidly.

However, despite what is claimed in the letter of answers by the corresponding author, the suggestions on the statistical methods to improve the analysis of results has not been really considered. Therefore, I suggest that the authors take time to contact a biometrician to apply a repeated measure analysis of variance to analyse the differences in log transformed epg values between the animals receiving or not pomegranate macerates and also, to have an advice on the best statistical method to apply for the results of larval cultures between groups.

Author Response

The authors are particularly grateful for spotting this side of the manuscript that certainly needed an improvement, to meliorate the quality of the manuscript.

Now, Fishers Exact tests were used for analysis of differences in the percentages of GINs genera in the coprocultures, found pre- and post-treatment, in the different studies groups.

The dataset concerning EPG valued to assess the efficacy of the treatments has now been analysed by a biometrician that applied the ANOVA, as suggested by the reviewer, followed by the student’s t test for each-pair comparison.

The amendments are described in the paragraph 2.5 (Methods>Statistical analysis) (lines 251-258).

The coproculture tables and all the results section have now been amended accordingly, adjusting the p-values and some description of the results/figures.

Reviewer 2 Report

Dear Author

Thanks for correct paper. The alteractions are verificated.

Author Response

Thank you very much for considering our manuscript and many thanks for the comments that helped improve the paper.